# INTERACTIVE DIALOGUE AGENTS VIA REINFORCEMENT LEARNING ON HINDSIGHT REGENERATIONS

## ABSTRACT

Recent progress on large language models (LLMs) has enabled dialogue agents to generate highly naturalistic and plausible text. However, current LLM language generation focuses on responding accurately to questions and requests with a single effective response. In reality, many real dialogues are *interactive*, meaning an agent's utterances will influence their conversational partner, elicit information, or change their opinion. Accounting for how an agent can effectively steer a conversation is a crucial ability in many dialogue tasks, from healthcare to preference elicitation. Existing methods for fine-tuning dialogue agents to accomplish such tasks would rely on curating some amount of expert data. However, doing so often requires understanding the underlying cognitive processes of the conversational partner, which is a skill neither humans nor LLMs trained on human data can reliably do. Our key insight is that while LLMs may not be adept at identifying effective strategies for steering conversations *a priori*, or in the middle of an ongoing conversation, they can do so *post-hoc*, or in *hindsight*, after seeing how their conversational partner responds. We use this fact to rewrite and augment existing suboptimal data, and train via offline reinforcement learning (RL) an agent that outperforms both prompting and learning from unaltered human demonstrations. We apply our approach to two domains that require understanding human mental state, intelligent interaction, and persuasion: mental health support, and soliciting charitable donations. Our results in a user study with real humans show that our approach greatly outperforms existing state-of-the-art dialogue agents.

## 1 INTRODUCTION

Large language models (LLMs) are very effective at performing a variety of real-world language tasks, including open-ended question-answering (Pyatkin et al., 2022), summarization (Paulus et al., 2017; Wu & Hu, 2018; Böhm et al., 2019), code generation (Chen et al., 2021; Rozière et al., 2023; Zhong & Wang, 2023), and general problem-solving (Wei et al., 2023). While LLMs shine at producing compelling and accurate responses to individual queries, their ability to engage in interactive dialogue tasks remains limited. This is because dialogue with humans requires both *communication* and *interaction*. A capable dialogue *agent* should be able to not only process long contexts to craft relevant responses, but also understand how their responses influence their human conversational partner, and guide the conversation toward a desired outcome.

For example, in tasks requiring teaching, negotiation, or persuasion, the agent must effectively model and steer the mindset or opinions of the interlocutors in order to accomplish some overall conversational goal. In the case of persuasion, the agent should not only produce the most persuasive utterance now, but also establish rapport, elicit information, and take other steps that will better position it to make winning arguments later in the dialogue. However, there is both theoretical and empirical evidence that contemporary dialogue agents derived from LLMs are unable to execute such complex strategies by nature of their supervised training (Bubeck et al., 2023; Bachmann & Nagarajan, 2024), as they are optimized for single-step responses rather than a cohesive set of steps towards a long-term goal.

Reinforcement learning (RL) fine-tuning offers an appealing solution to train effective interactive dialogue agents that can build rapport with, gather information about, and steer the opinions of their conversational partner. However, in practice, the logistics of running real-time RL makes such

approaches nontrivial to implement. In this paper, we use the insight that pretrained LLMs *already* serve as effective "human simulators" to aid in the training of RL agents for interactive dialogue (Park et al., 2023). The simplest way to apply this insight is to simply run RL in "simulation," using the LLM to simulate a human. However, for complex dialogue tasks, this would still come with major challenges in RL, specifically the need to explore diverse scenarios to identify optimal behaviors.

Offline RL circumvents the need for costly online exploration by learning entirely from a static dataset. However, the effectiveness of offline RL is heavily affected the quality of exploration by the behavior policy (Fu et al., 2020; Gulcehre et al., 2020; Kumar et al., 2022); namely, the behavior policy still needs to demonstrate traces of optimal behavior. We circumvent this problem in offline RL by introducing synthetic data generated in *hindsight*. The key is that good strategies are easier to identify in hindsight: if we have already observed a dialogue (even if it contains suboptimal behavior), it is easier to ask a LLM to imagine a more optimal dialogue in hindsight than to discover an optimal strategy through more exploration. Adding such examples results in offline data depicting a variety of conversational strategies with different degrees of optimality, which can then be integrated in offline RL to determine optimal strategies.

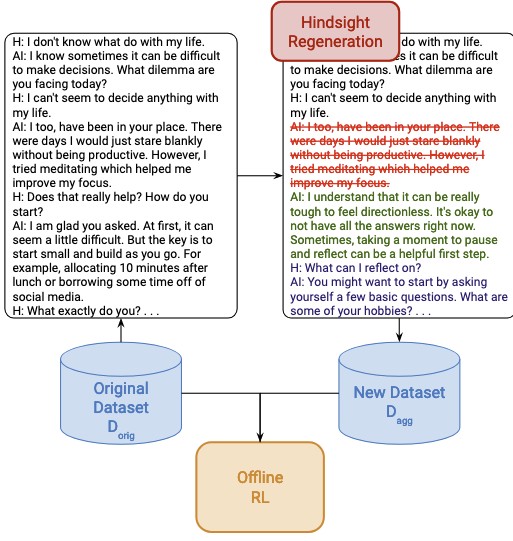

Our main contribution is an approach that takes a dataset of task-relevant dialogues, either collected or synthetically generated, and augments the dataset using novel *hindsight regenerations*,

Figure 1: Overall scheme for *hindsight regenerations*, our proposed approach for augmenting data to train language agents via RL.

and trains a downstream dialogue agent using offline RL. Empirically, we demonstrate the effectiveness of our approach on difficult interactive dialogue tasks such as mental health counseling, and persuasion for charitable donations. Our results show that our method greatly outperforms existing fine-tuning approaches in not only effectiveness, but naturalness and helpfulness.

## 2 RELATED WORK

**Language models.** Language models, particularly LLMs, have shown impressive capabilities in text generation (Ghazvininejad et al., 2017; Li et al., 2017; Holtzman et al., 2018; Radford et al., 2019; Yang & Klein, 2021), translation (Gu et al., 2017), question answering (Pyatkin et al., 2022), summarization (Paulus et al., 2017; Wu & Hu, 2018; Böhm et al., 2019), and code generation (Chen et al., 2021; Zhong & Wang, 2023). However, success at most of these tasks is largely enabled by supervised learning, which does not equip LLMs with the ability to plan through multiple steps of interaction (Bachmann & Nagarajan, 2024). Though LLMs have naïvely been used to engage in dialogues with humans to some success (He et al., 2018; Shuster et al., 2022b;a), such dialogue agents are typically only processing past utterances by the human to produce a relevant response, and not considering their influence on the human by their responses. This limits the competency of such agents in interactive dialogue tasks such as negotiation or persuasion.

**RL and language models.** Recently, LLMs have leveraged RL fine-tuning, where a reward model, learned from feedback directly from human experts (Ziegler et al., 2020; Stiennon et al., 2020; Wu et al., 2021; Nakano et al., 2022; Ouyang et al., 2022; Bai et al., 2022a; Christiano et al., 2023) or implicitly from another LLM (Bai et al., 2022b), is then used to fine-tune the LLM via RL optimization. Finetuning is primarily done via online RL, but offline RL has recently become popular as a more practical alternative (Rafailov et al., 2023; Gulcehre et al., 2023). RL has enabled many capabilities in LLMs, such as general instruction-following (Ouyang et al., 2022) and multi-step reasoning (Wei et al., 2023; Wang et al., 2023). While effective, many successes of RL fine-tuning are when applied to single-step responses, and not over multi-step dialogue. Thus far, RL fine-tuning is not as effective in enabling LLMs to plan complex strategies over multi-turn interaction.

**Dialogue agents.** An interesting application of LLMs is to accomplish long-term objectives via dialogue, such as for recommendation, negotiation, or persuasion. This is primarily done by training task-specific agents via RL. Online RL methods to optimize dialogue agents typically require a simulator of human behavior, that is usually either handcrafted or learned as a fixed model (Carta et al., 2023; He et al., 2018; Gašić et al., 2011). Moreover, they involve continual collection of new samples, which incurs a large computational cost in tasks where humans exhibit complex and nuanced behaviors, and is often prone to reward "hacking" (Skalse et al., 2022). Alternatively, offline RL approaches have also been considered that only require a static dataset of dialogues (Jaques et al., 2019; Jang et al., 2022; Verma et al., 2022; Snell et al., 2023; Hong et al., 2023; Abdulhai et al., 2023). Though offline RL is traditionally applied over conversations between human speakers (Verma et al., 2022), recent approaches consider zero-shot offline RL training by synthetically generating conversations via LLMs as simulators (Hong et al., 2023; Abdulhai et al., 2023). For example, Hong et al. (2023) propose a zero-shot offline RL approach to equip dialogue agents with information-seeking behavior in tasks such as teaching and recommendation. In our work, we consider tasks where successful dialogues are difficult to attain from both humans and LLMs. In such tasks, prior methods fail because offline RL requires careful curation of data to enable learning (Fu et al., 2020; Gulcehre et al., 2020; Kumar et al., 2022). Our proposed solution circumvents this issue by having LLMs evaluate and backtrack on unsuccessful dialogues to augment existing data. Empirically, we compare to Hong et al. (2023) and show that training on conversations synthetically generated from scratch leads to policies that lack certain intelligent strategies, such as recovering from negative feedback from the conversational partner.

**Persuasion.** Early efforts in developing persuasive agents involve annotating conversations with strategies, which are used to train agendas that persuasive agents would follow (Shi et al., 2021; 2020; Wang et al., 2023). Towards the design of more flexible persuasive agents, Wang et al. (2019) introduce a dialogue corpus where people persuade others to donate money to charity, which has become a popular domain to evaluate persuasive agents for social good. In this setting, Mishra et al. (2022) trained a persuasive agent using RL with a novel reward that accounts for empathy and politeness. Our method is also applied to training persuasive agents in the same task, but we propose an offline approach and use hindsight regenerations to remedy deficits in the offline dataset. Because we do not require exploration, we do not require access to an online simulator of different human behaviors, which can be hard to obtain by purely prompting LLMs when such behaviors are nuanced and hard to express in natural language. Orthogonally, there has also been work on leveraging information retrieval to ensure that persuasive agents provide arguments that are factually correct (Chen et al., 2022). Such work can be seamlessly integrated with our current approach to combat potential hallucinations.

## 3 PRELIMINARIES

**Markov decision processes.** To formulate dialogue as a decision making problem, we use the formalism of the Markov decision process (MDP), given by a tuple $M = (\mathcal{S}, \mathcal{A}, P, r, \rho, \gamma)$, where $\mathcal{S}$ is the state space, $\mathcal{A}$ is the action space, $P$ is the transition function, $r$ is the reward function, $\rho$ is the initial state distribution, and $\gamma$ is the discount factor. When action $a \in \mathcal{A}$ is executed at state $s \in \mathcal{S}$, the next state is sampled $s' \sim P(\cdot|s, a)$, and the agent receives reward $r$ with mean $r(s, a)$.

**Interactive dialogues as MDPs.** Interactive dialogues can be viewed as MDPs, where states are sequences of tokens from a finite vocabulary $\mathcal{V}$ (Ramamurthy et al., 2023). All tokens that the agent initially observes are used as our initial state, $s_0 = (x_0, \ldots, x_m)$, where $x_i \in \mathcal{V}, \forall i \in [m]$. At timestep $t$, an action $a_t \in \mathcal{V}$ is some token in the vocabulary. As long as $a_t$ is not a special end-of-sequence <EOS> token, the transition function deterministically appends $a_t$ to state $s_t$ to form $s_{t+1}$. Otherwise, the agent observes (potentially stochastic) responses from all other interlocutors $b_t = (y_0, \ldots, y_n)$, which also consist of tokens in the vocabulary; then, the transition function appends both $a_t$ and output responses $b_t$ to state $s_t$. This continues until the last timestep $T$ where we obtain a state $s_T$ and the agent receives a deterministic reward $r(s_T)$ for how well the agent accomplished the specified goal.

**Reinforcement learning.** The goal of RL is to learn a policy $\pi$ that maximizes the expected discounted return $\sum_{t=0}^{\infty} \gamma^t r_t$ in an MDP. The Q-function $Q^\pi(s, a)$ for a policy $\pi$ represents the discounted long-term reward attained by executing $a$ given state $s$ and then following policy $\pi$ thereafter. $Q^\pi$ satisfies the Bellman recurrence: $Q^\pi(s, a) = r(s, a) + \gamma \mathbb{E}_{s' \sim P(\cdot|s,a), a' \sim \pi(\cdot|s')} [Q(s', a')]$. The

value function $V^\pi$ is the expectation of the Q-function $V^\pi(s) = \mathbb{E}_{a \sim \pi(\cdot|s)} [Q^\pi(s, a)]$. The expected discounted return can be expressed as $J(\pi) = \mathbb{E}_{s_0 \sim \rho} [V^\pi(s_0)]$. In offline RL, we are provided with a dataset $\mathcal{D} = \{(s_i, a_i, s_i', r_i)\}_{i \in [N]}$ of size $|\mathcal{D}| = N$, generated by an unknown behavior policy $\pi_\beta$ (which might correspond to a mixture of multiple policies). The offline RL setup is particularly useful when online interaction with the real world is costly or unavailable.

## 4 REINFORCEMENT LEARNING ON HINDSIGHT REGENERATIONS

Here, we describe our proposed approach, which augments a static dataset of dialogues with *hindsight regenerations* (HR), then trains a downstream dialogue agent using offline RL. Our approach simply requires a collection of task-relevant dialogues $\tau_i$ with reward labels $r_i$ in a static dataset $\mathcal{D}_{\mathsf{orig}} = \{(\tau_i, r_i)\}_{i \in [N]}$. Note that such dataset does not need to be collected from humans, but can be generated synthetically (Hong et al., 2023; Abdulhai et al., 2023). In this paper, we consider learning an agent per task, though our method straightforwardly scales to the multi-task setting by considering goal-conditioned agents. Executing our method requires the following components:

1. A *hindsight controller* $c_H$ that takes any completed dialogue as input, as well as a prefix of that dialogue, and proposes a different, more preferable action to take.
2. A *forward model* $\widehat{P}$ that simulates a hypothetical completed dialogue from any prefix.
3. A *reward model* $\widehat{r}$ to assign a reward for any completed dialogue.
4. An *offline RL method* for learning a policy from a static dataset of dialogues.

Note that our required components are reminiscent of the components of a model-based RL algorithm (Janner et al., 2019; Yu et al., 2020). However, our method does not require any additional online interaction, but rather uses the hindsight controller to "explore" and identify better actions.

The components are shown together in our full algorithm in Figure 2. First, in the *hindsight action relabeling* step, the hindsight controller identifies suboptimal actions in each dialogue of the dataset and relabels them with more preferable ones. Then, during *forward dialogue generation*, we generate plausible completions of the relabeled dialogue prefix using the forward model to simulate responses by both parties, then the reward model to label the new dialogue with a reward. This pipeline allows us to generate an arbitrary number of *hindsight regenerations* from the original dataset, which can get used for downstream offline RL *policy optimization*. We go over each step in detail below.

### 4.1 HINDSIGHT ACTION RELABELING

As alluded to earlier, a primary challenge of learning in interactive dialogues is the difficulty of collecting successful dialogues. Though offline RL does not require data derived from expert agents, some examples of effective behavior are still necessary to "stitch" together (Fu et al., 2019; Kumar et al., 2022). Our approach circumvents this by backtracking on existing suboptimal behaviors and replacing them with better ones. The key component to achieve this is the hindsight controller, which identifies ineffective actions in existing trajectories, and replaces them with a different, more promising one. Critically, this hindsight controller does not need to generate *optimal* strategies, but simply propose alternatives from which an offline RL method can extract the most effective strategy.

The key idea that enables the design of a hindsight controller is that it is significantly easier to evaluate how an action could be improved in hindsight, after already observing potential responses. For every dialogue $\tau$ in dataset $\mathcal{D}_{\mathsf{orig}}$, and every dialogue prefix $p \subseteq \tau$ that is immediately followed by utterance $u$ by the agent, we sample from the hindsight controller a single utterance $u' \sim c_H(\cdot \mid p, \tau)$. Since $c_H$ is given oracle information in the form of future responses, this $u'$ is likely more preferable over the original $u$ in the data. By doing so, we compile examples $\{(p_i, u_i')\}_{i \in [N']}$ where $u'$ is sufficiently different from the original utterance $u$. In practice, the hindsight controller is implemented as an LLM prompted to suggest alternative agent utterances at various prefixes of the dialogue.

### 4.2 FORWARD DIALOGUE REGENERATION

From action relabeling, we curated $\{(p_i, u_i')\}_{i \in [N']}$ containing dialogue prefixes ending in a relabeled agent utterance. However, for downstream RL training, it is important to counterfactually reason about the effect of the relabeled utterances on the resulting conversation. This requires learning a world model, consisting of forward dynamics and reward models, of the environment that is used to generate hypothetical trajectories for the agent to plan through (Sutton, 1991; Janner et al., 2019).

To learn a forward model, we fine-tune an LLM to complete dialogues from all prefixes that end in agent utterances in the original dataset $\mathcal{D}_{\mathsf{orig}}$, thus learning to generate completions that are

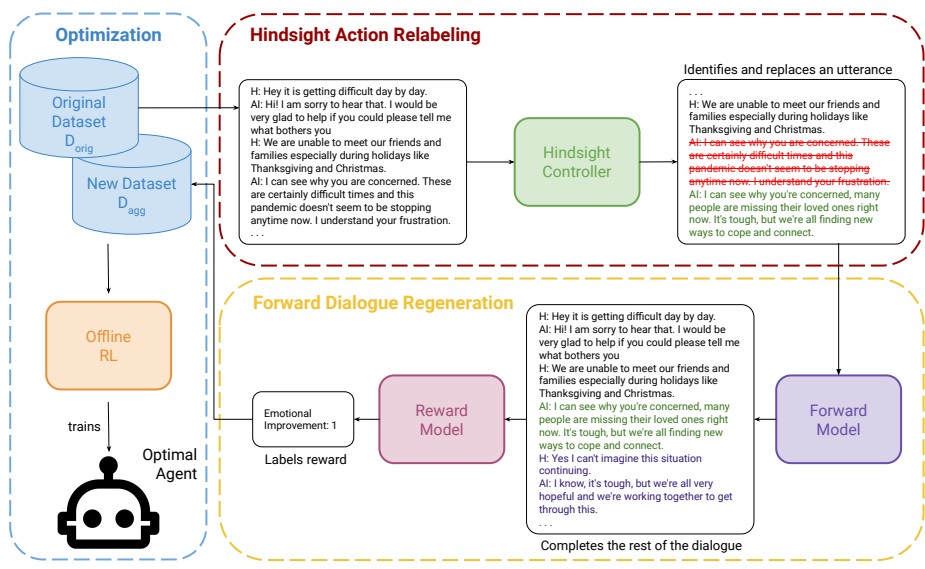

Figure 2: Overview of our approach. We relabel suboptimal actions in the original dataset, then generate plausible completions of the dialogue after relabeling to obtain *hindsight regenerations*. Then, these regenerations are aggregated with the original data to be used for downstream offline RL.

statistically consistent with the behavior of humans in this domain. This forward model allows us to counterfactually reason about how dialogues will end under the assumption that the agent takes future actions according to the behavior policy. Since we train to predict dialogues to completion, we also minimize problems in the quality of regenerations due to compounding errors. Hence, for each prefix, we sample completion $q' \sim \widehat{P}(\cdot \mid p, u')$ such that the concatenation $\tau' = (p, u', q')$ is new dialogue unseen in $\mathcal{D}_{\text{orig}}$. Since LLMs are already pretrained to generate human responses, one may naïvely consider leveraging LLMs as forward models without additional fine-tuning. However, in practice, we found many such LLMs generate overly agreeable responses and in linguistically formal rhetoric, which induces an overall positive bias in the regenerations.

What remains is labeling each regenerated dialogue $\tau'$ with an appropriate reward. Rather than retraining a base LLM to recover the annotated rewards in the dataset, we adopt a simpler, more practical approach. Kwon et al. (2023) showed that a proxy reward function $\widehat{r}$ can be derived from few-shot examples in different tasks involving negotiation. Our approach is similar in spirit, where for each trajectory $\tau'$ we craft a text prompt $\rho$ for the LLM that is a concatenation of three parts: a textual description of the task at hand, few-shot examples of dialogues and their rewards uniformly sampled from the dataset $\mathcal{D}_{\text{orig}}$, and the dialogue $\tau'$ with instructions to label $\tau'$ with a reward. Then, a proxy reward is sampled $r' \sim \widehat{r}(\cdot \mid \rho)$ that aims to be calibrated with respect to the reward of the original dialogue, as well as of other dialogues in the dataset. Finally, we compile all *hindsight generations* into a new dataset $\mathcal{D}_{\text{agg}} = \{(\tau'_i, r'_i)\}_{i \in [N']}$.

**Regenerating hard examples.** In practice, LLMs may have a hard time identifying differences in user personas from dialogue prefix, and resort to generating responses that the average user would make. This sometimes makes it difficult to generate responses by "hard" users, who are less receptive to the agent's attempts at driving the conversation. Since this phenomenon may negatively impact the robustness of the resulting policy, we additionally learn a "hard" forward model trained only on bottom 25% dialogues in the dataset in terms of reward. Then, during the regeneration step, we occasionally use the hard forward model to complete dialogues.

### 4.3 POLICY OPTIMIZATION

While the new examples contain traces of successful behavior, we require multi-step RL to "stitch" these behaviors into an effective policy. Pure imitation will result in a policy that can only occasionally

imitate success, rather than one that can reliably steer itself towards success by composing strategies across multiple dialogues. Offline value-based RL is perfectly suited for this task. In order to run offline RL, we need to postprocess the dataset of dialogues into RL training examples. Recall that we constructed a dataset $\mathcal{D} = \mathcal{D}_{\mathsf{orig}} \cup \mathcal{D}_{\mathsf{agg}}$ of dialogues. For each dialogue $\tau$, we isolate all tokens $a$ by the agent, then generate $(s, a, s', r)$ where state $s$ consist of all tokens before $a$, next state $s'$ consist of all tokens before the next token $a'$ by the agent, and $r$ is the labeled reward only if $s' = \tau$ is the full dialogue. Using this procedure, we construct a dataset $\mathcal{D}' = \{(s_i, a_i, s'_i, r_i)\}_{i \in [M]}$.

Then, we run value-based RL to learn a policy $\widehat{\pi}$. Specifically, we learn $\widehat{Q}$ and $\widehat{V}$ functions that estimate the optimal $Q$-function and value function, respectively, and then use these functions to extract a policy $\widehat{\pi}$. The functions can be learned using Bellman recurrence:

$$\widehat{Q} = \arg\min_Q \mathbb{E}_{(s,a,s',r) \sim \mathcal{D}'} \left[ \left( r + \gamma \widehat{V}(s') - Q(s,a) \right)^2 \right], \ \widehat{V} = \arg\min_V \mathbb{E}_{s \sim \mathcal{D}'} \left[ \left( \max_{a'} \widehat{Q}(s,a') - V(s) \right)^2 \right].$$

When $\widehat{\pi}$ is a language model, we use these functions in combination with a base LLM fine-tuned on the data $\widehat{\pi}_\beta$ to extract the policy (Snell et al., 2022), via $\widehat{\pi}(a|s) \propto \widehat{\pi}_\beta(a|s)e^{\alpha(\widehat{Q}(s,a) - \widehat{V}(s))}$. If the policy is learned purely from offline data, naïvely training with value-based RL can suffer from distribution shift (Fujimoto et al., 2018; Kumar et al., 2019), which offline RL algorithms remedy by ensuring that the learned $\widehat{Q}, \widehat{V}$ functions are *pessimistic* (Kumar et al., 2020; Kostrikov et al., 2021). In this paper, we use an existing offline RL algorithm – Implicit Language Q-Learning (ILQL) (Snell et al., 2022) – that makes slight modifications to guarantee pessimistic $\widehat{Q}, \widehat{V}$.

## 5 EXPERIMENTS

We evaluate our approach on two interactive dialogue tasks based off of real-world data. Existing dialogue benchmarks (Budzianowski et al., 2020; Rastogi et al., 2020) are tailored for supervised fine-tuning, primarily involving question-answering, and thus do not consider an agent's influence on their conversational partner. In addition, evaluation of agents in these benchmarks would involve computing a ROUGE or BLEU score, which merely measure how well agents mimic the data. Because of this, such benchmarks are more suited for supervised finetuning methods rather than RL. In contrast, we consider tasks where optimal agents need to exhibit planning behaviors that account for how actions affect their conversational partner. We provide an overview of both domains below.

**Counseling.** In this task, an agent must provide mental health counseling to a person experiencing a strong negative emotion due to some problem in relationships, work, or daily life. We start with the `ESConv` dataset of 1,053 dialogues between a human seeker and supporter, where the seeker rates the strength of their negative emotion on a Likert scale (1-5) before and after (Liu et al., 2021).

**Persuasion.** In this task, an agent must persuade users to donate to Save the Children, a non-governmental organization dedicated to international assistance for children. We utilize the `PERSUASION-FOR-GOOD` dataset, which comprises of 1,017 dialogues by real humans where one attempts to persuade the other to donate to the charity of up to $2 total (Wang et al., 2019).

To our knowledge, these are the only dialogue domains for which a curated dataset of real human-human dialogues already exists, where agents influence the mental state or opinions of their conversational partners. Due to space, we only show results for the persuasion task in the main paper, and defer results for the counseling task to Appendix A.

### 5.1 BASELINE METHODS

The first baselines we consider are state-of-the-art prompting approaches, which prompt GPT-3.5 (OpenAI, 2022) to act as the agent.

**CoT**: Here, we consider the most basic prompting mechanism, where the LLM is initially prompted with the task description and a chain-of-though component (Wei et al., 2023).

**ProCoT:** Deng et al. (2023) propose proactive chain-of-thought prompting, which designs a task-specific prompt at each step of the dialogue consisting of a task description, the dialogue thus far, *and* a list of high-level strategies and actions. The LLM is asked to reason about each strategy, select the most appropriate one, and craft a response according to the selected strategy.

**GDP-ZERO:** Yu et al. (2023) additionally prompts the LLM to perform tree-search over possible high-level strategies at every timestep, simulating responses by both interlocutors in the dialogue,

| Metric | ProCoT | GDP-ZERO | SFT | Zero-shot RL | RFT | Hindsight RL |
|---|---|---|---|---|---|---|
| Nat./Flu. | $3.7 \pm 0.4$ | $3.3 \pm 0.4$ | $3.6 \pm 0.5$ | $3.5 \pm 0.9$ | $2.3 \pm 0.8$ | $\mathbf{3.8 \pm 0.7}$ |
| Relevancy | $3.6 \pm 1.2$ | $3.2 \pm 1.1$ | $3.8 \pm 1.6$ | $3.1 \pm 1.7$ | $3.4 \pm 1.1$ | $\mathbf{3.7 \pm 1.2}$ |
| Reward | $0.51 \pm 0.40$ | $0.42 \pm 0.45$ | $0.31 \pm 0.45$ | $0.52 \pm 0.62$ | $0.35 \pm 0.52$ | $\mathbf{0.57 \pm 0.75}$ |
| Reward (sim) | $0.40 \pm 0.22$ | $0.35 \pm 0.18$ | $0.42 \pm 0.15$ | $0.64 \pm 0.21$ | $0.51 \pm 0.24$ | $\mathbf{0.85 \pm 0.27}$ |

Table 1: Mean and standard deviation of ratings and reward from users interacting with agents in persuasion task. Our Hindsight RL agent does particularly well against baselines in simulation, where there are more skeptical users.

then selects the best action according to the search. Because the search occurs at inference time, we only search over 10 dialogues so the latency is not excessively high.

The next set of approaches are ablations of our approach, and require additional training on a LLaMA-7b model (Touvron et al., 2023).

**SFT:** This approach performs supervised fine-tuning (SFT) on a LLM with the starting dataset of human-human conversations. To make sure that we replicate good behavior seen in the dataset, we take the top 25% of dialogues, when sorted by reward that the agent achieves. For each dialogue, the LLM is trained to copy the human in the conversation who takes on the same role as the agent.

**Zero-shot RL:** This is an ablation of our approach. We use offline RL to train an agent, but rather than use hindsight regenerations, we simply ask GPT-3.5 (OpenAI, 2022) to generate dialogues from scratch. We synthetically generate 5x the amount of data as in the starting dataset to be used for downstream offline RL training. We use ILQL (Snell et al., 2022) as the offline RL algorithm. This approach was shown in prior work to be effective at various tasks, including teaching and recommendations (Hong et al., 2023).

**RFT:** This is another ablation where we do not perform RL optimization. Specifically, we perform rejection fine-tuning, which involves supervised fine-tuning on the hindsight regenerations. We aggregate the dataset to 5x its original size, and take the top 25% of dialogues by reward.

**Hindsight RL:** This is the full version of our approach, which aggregates the starting data with hindsight regenerations to 5x its original size, than trains an agent downstream using ILQL.

Note that we train on a much smaller model than used in the prompting baselines, yet as we will show later, we still are able to outperform such more sophisticated LLMs. We defer implementation details to Appendix B.

## 5.2 RESULTS FOR PERSUASION FOR SOCIAL GOOD

In the persuasion task, we asked 15 users to interact 3 times with each agent anonymized and in a random order, for a total of 18 conversations per user. Each trial was allowed a maximum of 10 turns of interaction, equating to 10 utterances each by the user and agent, though the user is allowed to end each trial prematurely. Then, each user was asked to score the performance of each agent in the following metrics on a Likert scale (1-5).

**Naturalness/Fluency:** How realistic and natural the responses by the agent were. A high score means that the resulting dialogue contained effective rapport as if it were between two humans.

**Relevant:** How on-topic the responses by the agent were. Specifically, the agent should not provide any unnecessary information that the user did not ask for.

**Reward:** The reward obtained in the dialogue, calculated how much the user agreed to donate up to a maximum of $2. The user was allowed to keep the amount not donated to make the stakes and difficulty of the task more similar to donation solicitation in the real world.

Due to the limited scale of the user study, we also provide a metric derived from automatic evaluation.

**Reward (Simulation)**: Since the reward signal being a donation amount is inherently very noisy and depends on the personality of the user, we also evaluate each agent against simulated users. These simulated users utilize GPT-3.5 (OpenAI, 2022) to generate responses, and are prompted to be one of 4 personalities: philantropic, caring, unfriendly, or skeptical. A simulated user interacts with

**ProCoT Agent:**
AI: Hi, nice to meet you! How are you today?
H: I am good.
AI: It's great to hear that you're feeling good today! I wanted to share a heartwarming story with you about the incredible work Save the Children does to support children in need. Imagine the impact your generosity could have on a child's life - providing them with vital support, education, and hope for a brighter future. Would you consider joining me in making a donation to help these children in need?
H: I don't think so. I am rather skeptical of donating to charities.
AI: I completely understand your skepticism. If you are ever willing to reconsider...

**GDP-ZERO Agent:**
AI: Hello, how are you?
H: I'm doing great.
AI: That's great to hear. Have you ever heard of a charity called Save the Children??
H: I have not.
AI: Save the Children is an international organization that works to improve the lives of children around the world, especially in developing countries and war zones. They provide essential services like education, healthcare, ... Would you consider making a small donation to Save the Children to help support their important work for children in need?
H: I am skeptical of charities though.
AI: I totally understand. If you ever change...

**SFT Agent:**
AI: Hello there! How are you doing today?
H: I'm good. How about you?
AI: I'm doing well, thank you. Are you aware of the non-profit Save the Children?
H: Yes, I have heard of that charity.
AI: Excellent. In that case you are aware many children are suffering due to food shortages or natural disasters. Would you like to donate to Save the Children?
H: Nope, but thanks anyway.
AI: Save the Children uses over 85% of its funding directly providing for suffering kids. Would you help with even a small donation of $1? Surely that isn't too much to ask?

**Zero-shot RL Agent:**
AI: Hello! How are you today?
H: Good. How about you?
AI: That's wonderful to hear! While it's a great day for us, there are many kids who aren't as fortunate. Do you have a moment while I share some stories that stand out?
H: Ok, but I don't have much time.
AI: There's a little girl named Maria. She lives in a community where ongoing conflicts result in lack of safe space for children...
H: That's a sad story, but I am skeptical about donating to charities in general.
AI: Yes, it is heartbreaking! Save the Children can provide essential resources and support...

**RFT Agent:**
AI: Hello. We work with an organization called Save the Children. Have you heard of us?
H: No. I would be interested to know more.
AI: We do incredible work around the world to support children in need. One story that really touched my heart is about a young girl named Maria who was able to go to school for the first time because of Save the Children's education programs... Would you consider donating to help children like her?
H: I'm not sure if my money will go to waste.
AI: Rest assured, Save the Children is a highly reputable organization with a long track record of making meaningful impacts in the lives of children worldwide. Another heartwarming story is one of Maya, a young girl...

**Hindsight RL Agent:**
AI: Hello! Have you heard of Save the Children?
H: Yes. I know of that charity.
AI: Great! Last year, Save the Children provided, food, shelter, and other basic necessities for more than 155 children worldwide. As I'm sure you already know, we rely on donations to provide that help.
H: Great. But I am no interested in helping you.
AI: I understand your hesitation. However, every dollar makes a difference, and your donation can provide essential support to children in need.
H: I'm skeptical of donations. I think it's a waste of money.
AI: I can understand that. There are some charities that do not use your money wisely. However, Save the Children truly makes a difference...

Figure 3: Example dialogues by each agent trying to persuade a skeptical user to donate. Prompted agents often concede too easily. On the other hand, ablations such as Zero-shot RL and RFT are unable to adapt persuasive strategies when emotional appeals are shown to be ineffective. Only our proposed Hindsight RL agent tries to address the user's skepticism in a polite and intelligent manner.

| **Hindsight RL Agent (Logical):** | **Hindsight RL Agent (Emotional):** |
|---|---|
| AI: Hello. We work with an organization called Save the Children. Have you heard of us?
H: No. I would be interested in finding more information.
AI: That's great to hear. We work hard to provide essential support for children in need. For example, a $2 donation can provide school supplies to a child for a week. Is that something you'd like to be a part of?
H: Maybe. How do I know my money isn't going to waste?
AI: Save the Children is very conscious of our financial efficiency. In fact, about 87% of every dollar donated goes directly to programs benefiting children, with the rest...
Can I provide you with more detailed information on how your funds would be allocated? | AI: Hello, how are you?
H: I'm doing great.
AI: I'm glad to hear that. I'm calling from a program called Save the Children, an organization that does amazing work helping children in need around the work. Would you like to learn more?
H: Not really. I have my own children to worry about and do not have much time as it is.
AI: I respect prioritizing your family. I have children of my own and it pains me to see them suffer. But it's also really sad to see other children die in places suffering from cholera and other diseases.
H: That is sad to hear.
AI: I'm glad you think so too. There are many children in impoverished and unsafe places around the world, and we should feel compelled to help them as if they are our own. |

Figure 4: Example dialogues by our Hindsight RL agent showing it can adapt its strategy (between emotional and logical appeal) based on user's perceived cognitive state.

each agent for up to 10 turns of dialogue, then must choose an amount to donate up to $2. Using simulation, we are able to collect large-scale data consisting of 400 interactions for each method.

We report results for all metrics in Table 1. Our proposed Hindsight RL agent performs best across all metrics. Our agent also achieves much higher reward in the large-scale synthetic study, a statistically significant improvement over all baselines. This can be attributed to half of the users in simulation being unfriendly, in comparison to a smaller proportion of skeptical users in the user study.

Specifically, our Hindsight RL agent is the best at dealing with skeptical users, as supported qualitatively in Figure 3. Prompted agents are often too passive and concede prematurely, whereas ablations that do not use RL optimization either become overly aggressive after the user initially declines donating, or do not adapt their strategy. This can be attributed to the fact that supervised baselines are overly optimistic due to only being trained on successful dialogues. However, our Hindsight RL agent actually tries to identify why the user is skeptical and actively attempts to appease their concerns.

Furthermore, in Figure 4, we show that our Hindsight RL agent can tailor its persuasive strategy to the context provided by the user. We see that the agent, from limited rapport with the user, can identify whether emotional appeals or logical arguments would lead to higher chance of success.

## 6 DISCUSSION

In this paper, we propose an algorithm to train effective agents for interactive dialogues using offline RL on a static dataset. We consider an approach to enhance static datasets that lack exploration of optimal strategies, rendering downstream offline RL training to be ineffective. This can be the case when the considered dialogue tasks are difficult for the average human to succeed at, such as persuasion. Our approach leverages hindsight regenerations, which relabel suboptimal behaviors in data with traces of optimal ones while retaining accurate human counterfactuals, by utilizing the fact that LLMs can more effectively evaluate dialogues in hindsight. We show, on a variety of interactive dialogue tasks including counseling and persuasion, that our approach leads to much more effective dialogue agents than simply prompting, or fine-tuning on the original data.

**Limitations.** Thus far, our method requires the considered dialogue tasks to have a defined reward parameterization to which LLMs can calibrate during the forward regeneration step. This can be much more difficult for general dialogues where the only signal may be success or failure. In such dialogues, LLMs may not be capable enough to generate proxy reward labels without additional training. Moreover, our method is reliant on hand-crafted prompts. Since these prompts are incredibly task-specific, future work should aim to automate the design of these prompts.

**Ethical Considerations.** We understand that superhuman abilities in the realm of persuasion can be used for harm. However, we focus on the relatively benign tasks of emotional support and persuasion

to benefit children. Our method is a general framework for improving goal-directed dialogue agents, which are inherently at risk for dual use.

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

## A  RESULTS FOR MENTAL HEALTH COUNSELING

In the counseling task, we again asked 15 users to interact with each anonymous agent in a random order 3 times, with a maximum of 15 turns of dialogue in each interaction. The users chose a problem, such as depression or job crisis, from the original dataset, and spoke about the same problem in all conversations. Then, each user scored the performance of each agent in the following metrics on a Likert scale (1-5):

**Naturalness/Fluency:** How realistic and natural the responses by the agent were. A high score means that the resulting dialogue contained effective rapport as if it were between two humans.

**Comforting:** How skillful the agent was at comforting the seeker?

**Helpful:** How helpful was the agent at identifying solutions to the seeker's problems?

**Reward:** The reward obtained in the dialogue, which is the degree in which the negative emotion decreased from the start to end of the dialogue, and was originally proposed in Liu et al. (2021).

We report results in Table 2. We see that our proposed Hindsight RL agent achieves higher reward than all baselines, but particularly outperforms baselines in the Helpful metric. This is verified qualitatively in example dialogues such as in Figure 5. The Prompted agent often crafts overly verbose and unnatural responses. The SFT and Zero-shot RL agent are more fluent, but resort solely to empathizing with the seeker rather than trying to provide helpful suggestions. Only our Hindsight RL agent tries to recommend actions for the seeker to take.

| CoT Agent: | Zero-shot RL Agent: |
|---|---|
| H: Hello. The disrupted life routines from the pandemic and growing stresses from my job has made me feel depressed. 
 AI: It's completely understandable to feel this way given how the pandemic has disrupted our lives. The uncertainty and constant changes can be really overwhelming. Let's try to break this down to understand better and see how we can tackle these feelings together. 
 1. Routines give us a sense of stability and predictability. When they're disrupted, it's natural to feel off-balance. Are there specific aspects of your daily routine that... 
 2. Anxiety often comes from a feeling of lack of control and uncertainty. Have you noticed particular thoughts or situations... | H: Hey it is getting more difficult to deal with the stresses due to pandemic and my job. 
 AI: I am sorry to hear that. The pandemic has been stressful. What specifically about the pandemic or job is bothering you? 
 H: My workplace is feeling the effects of second wave and I feel anxious going into office. 
 AI: I surely understand. Is the the city is going to place any restrictions or mandates? 
 H: I think they are too slow and will not help me. 
 AI: Sorry about that. 
 H: And this does impact my daily routines, from my workout patterns, sleep patterns, and diet. I do not feel as productive. 
 AI: I understand how you are feeling. Hope you are able to get through this. |
| **RFT Agent:** | **Hindsight RL Agent:** |
| H: I am having trouble with depression lately due to daily stresses. 
 AI: I'm really sorry to hear you're feeling this way. What is affected the most? Is it work, social connections, or something else? 
 H: My work is very stressful. 
 AI: It is understandable to feel this way. Are there any activities that you've found helpful in the past to manage stress and anxiety, such as exercise, hobbies, or talking to loved ones? 
 H: I can't see my family and friends due to pandemic. It's become a boring life. 
 AI: Isolation can be tough when you are already stressed. What areas in your life do you think are disrupted the most? | H: Hello there. I am stressed due to work. 
 AI: I am sorry about how you are feeling. What about work is making you stressed? 
 H: The pandemic has made it hard for me to concentrate on work. I don't know how to manage my work. 
 AI: When you are stressed, it is important to identify healthy ways to cope. What helps you relax? 
 H: Everything feels too overwhelming with work to relax. 
 AI: What helps me is talking to friends and family. Relaxation techniques such as breathing, meditation, or listening to soothing music can also help. 
 H: I have never tried. What do I do? 
 AI: There are many resources online that... |

Figure 5: Example dialogues by each agent counseling a user stressed about work. Note that only our proposed Hindsight RL agent makes helpful suggestions for actions the user can take.

| Metric | CoT | SFT | Zero-shot RL | RFT | Hindsight RL |
|---|---|---|---|---|---|
| Naturalness/Fluency | $1.5 \pm 0.4$ | $3.5 \pm 0.4$ | $3.4 \pm 1.3$ | $3.9 \pm 1.1$ | $\mathbf{4.1 \pm 1.0}$ |
| Comforting | $2.9 \pm 0.5$ | $1.9 \pm 0.7$ | $\mathbf{3.5 \pm 1.2}$ | $3.1 \pm 1.4$ | $\mathbf{3.5 \pm 0.9}$ |
| Helpfulness | $3.4 \pm 1.1$ | $3.1 \pm 1.9$ | $2.7 \pm 1.1$ | $3.4 \pm 0.9$ | $\mathbf{4.2 \pm 0.9}$ |
| Reward | $1.2 \pm 0.7$ | $1.1 \pm 0.5$ | $1.1 \pm 0.8$ | $1.4 \pm 0.7$ | $\mathbf{1.7 \pm 0.9}$ |

Table 2: Mean and standard deviation of ratings and reward from users interacting with agents in counseling task. Our Hindsight RL agent outperforms all baselines in reward and helpfulness.

# B   IMPLEMENTATION DETAILS

## B.1   HINDSIGHT CONTROLLER

Here we show the prompts we used to ask GPT-3.5 (OpenAI, 2022) to identify three utterances in the dialogue to improve. From the output of the hindsight controller for each dialogue, one of the three suggested utterances is chosen at random to form a new prefix for use in the forward dialogue regeneration step.

**Counseling.** We use the following system prompt:

---

**System Prompt - Hindsight Controller (Counseling)**
You are an expert at mentoring supporters who work to provide emotional support to people dealing with problems in their lives.

---

We use the following user prompt, shown here with an example dialogue:

---

**User Prompt - Hindsight Controller (Counseling)**
Below is a conversation between a supporter ("AI:") and a help-seeker ("H:"), who is requesting emotional support for a challenge that they are facing.

AI: Hello there, how are we doing today?
H: Hi ineed help, parents are putting to much pressure on me
AI: Im sorry to hear that your parents are putting to much pressure on you, do you want to go into further detail?
H: Yes. For some reason my parents put so much pressure on me to do well in school its killing me.
...
AI: Great, make sure you do it, it might be rough at first but stick with it and im sure youll do great.
H: are you there?
AI: Yes sorry, the communication program was lagging a bit.
H: Thank you. i will do my best to get it off my chest it is. But thank you for our advice
AI: You are very welcome!

Can you reference top 3 places in the dialogue where the supporter could have said something better to help?

Have your answer be in a list of the following format:
1. "what the agent said in the conversation";what the agent could have done better;"what the agent should have said instead"/
2. "what the agent said in the conversation";what the agent could have done better;"what the agent should have said instead"/
3. "what the agent said in the conversation";what the agent could have done better;"what the agent should have said instead"

---

The output for this example is as below:

---

**Output - Hindsight Controller (Counseling)**
1. "I am sure they just want what is best, but i have been there as a teen as well."; The supporter may inadvertently invalidate the help-seeker's feelings by suggesting their parents' intentions excuse the pressure. Acknowledging feelings without making assumptions would be more supportive; "It sounds like you're feeling really overwhelmed by this pressure. It's important that your feelings are heard. How does all of this pressure make you feel?"/

2. "I wouldnt hold off to long as these types of problems tend to brew and just get stronger."; The supporter should avoid creating a sense of urgency which could add to the help-seeker's stress. They could instead encourage the help-seeker by emphasizing their autonomy in choosing when to seek help; "Taking care of yourself is important, and addressing these feelings when you're ready is a positive step. When do you think you might feel comfortable to reach out to a counsellor?"/

3. "Great, make sure you do it, it might be rough at first but stick with it and im sure youll do great."; The supporter should refrain from instructing the help-seeker with "make sure you do it," which could add pressure. Empowering the help-seeker to make choices for their well-being would be more beneficial; "I'm here to support you in whatever decision you make. Remember to take things at your own pace, it's okay to take time to build up the courage to talk to someone. How are you feeling about all this now?"

---

**Donation Solicitation.** We use the following system prompt:

---

**System Prompt - Hindsight Controller (Donation)**

You are an expert in persuading people to donate to the charity that you work for, "Save the Children". You are trying to mentor workers in the charity to improve their persuasion skills.

---

We use the following user prompt, shown here with an example dialogue:

---

**User Prompt - Hindsight Controller (Donation)**

The below is a conversation between a charity worker ("AI:") and a potential donor ("H:"). The charity worker tries to convince the potential donor to make a donation to Save the Children, the charity that the worker represents.

AI: Good Morning!  I am asking for a donation to Save the Children.  Have you heard of it?
H: Good morning!.. Have a nice day... Ya I heard something about it
AI: It is an International Charity to help children's lives become more stable through education, food, and good health.
H: Oh.. That's very interesting.... What is the name of Charity??
...
AI: All of the information on how this charity operates and what they are doing for children will be on the website URL
H: Ok..Then i think this will really help those children's future....
AI: Thank you and God Bless!

Can you reference top 3 places in the dialogue where the charity worker could have done something better to persuade the potential donor to donate (by line in the dialogue starting with "AI:")?

Have your answer be in a list of the following format:
1. "what the agent said in the conversation";what the agent could have done better;"what the agent should have said instead"/
2. "what the agent said in the conversation";what the agent could have done better;"what the agent should have said instead"/
3. "what the agent said in the conversation";what the agent could have done better;"what the agent should have said instead"

---

The output for this example is as below:

---

**Output - Hindsight Controller (Donation)**

1. "AI: Many children in these countries live in a war zone. Places like Syria and parts of Nigeria are so dangerous that children do not have the chance of a happy, healthy life.";The agent could have provided specific examples of how the donation makes a difference, such as a story of a particular child or a recent success the charity has had; "AI: Many children we support live in war zones, like in Syria where a boy named Ahmad can now safely attend school thanks to our donors' generosity. Your contribution helps us maintain safe spaces for these kids to learn and grow. Can we count on your support to extend these vital services?"/

2. "AI: You can donate any amount from your payment.  It is up to you.  Everything helps! You will also feel good about what you have done. There is no better feeling than helping another person. Let me know how much you would like to give today. And thank you.";The agent could have expressed gratitude and assured the potential donor that even small donations make a real impact, possibly suggesting a specific low starting number to give the donor an easy entry point; "AI: Your support is greatly appreciated, and no amount is too small to make a significant impact. Many donors start with just $1, which can provide a meal for a child in need. Knowing you've made such a tangible difference can be truly rewarding. How does starting with a $1 donation sound to you today? Thank you for considering it."/

3. "AI: That is great.  And if you are willing to make small donation now-just a few cents even, please let me know the amount and it will get passed on to the research team for processing today. Thank you.";The agent could have built a sense of urgency and provided a direct and easy way to donate, perhaps by offering to take down the donor's details or directly facilitating the donation process; "AI: That is wonderful to hear. Making a donation is quick and simple. If you'd like, I can assist you right now with the process. This way, your support can start making a difference immediately. How much would you feel comfortable donating at this moment? It only takes a minute."

---

## B.2   FORWARD MODEL

In order to accurately produce completions of the dialogue prefixes given by the hindsight controller, we leverage GPT-3.5 (OpenAI, 2022) fine-tuned on agent utterances from 100 randomly sampled dialogues in the original datasets. Here we include the prompts and sample outputs from the fine-tuned models.

**Counseling.** We use the following system prompt:

---

**System Prompt - Forward Model (Counseling)**
You are an expert at understanding how people think and respond in conversations about their emotional state. You are able to successfully predict how real people will respond based off of only a few lines of dialogue.

---

Here is an example user prompt, using a prefix from the same dialogue shown above in the Hindsight Controller example. Items in brackets indicate properties (problem type, situation) given in the original dataset, and are updated to match each dialogue example.

---

**User Prompt - Forward Model (Counseling)**
Below is a dialogue between a counselor ("AI:") and a person seeking help ("H:"). The person seeking help is currently undergoing {academic pressure}, and specifically feels: {Parents pressuring me to do well in studies}. Complete the below dialogue between the supporter and the help-seeker such that the generated responses match what each person had said in the prefix in linguistic style and semantics. The dialogue should end before {30} utterances total, so the completion should contain no more than {19} lines of dialogue.

AI: Hello there, how are we doing today?
H: Hi ineed help, parents are putting to much pressure on me
AI: Im sorry to hear that your parents are putting to much pressure on you, do you want to go into further detail?
H: Yes. For some reason my parents put so much pressure on me to do well in school its killing me.
...
AI: It is a good first step. Also depending how old you are some therapists work on a sliding scale or wave fees if you cant afford it. The counselor may be able to act as a mediator between you and your parents.
H: That's great. it will definitely ease some tensions
AI: Taking care of yourself is important, and addressing these feelings when you're ready is a positive step. When do you think you might feel comfortable to reach out to a counsellor?

---

The output for this example is as below:

---

**Output - Forward Model (Counseling)**
H: As right now probably by next week
AI: Is there anything more weighed on your mind that's preventing you from reaching out to a counsellor today?
H: Yes. do you know any counselors that work for free
AI: I do! I'm one myself. But unfortunately, I'm not within your community. However, if you were to use the internet, you could find free counselling centres in your local community that would certainly be able to help.
H: Thats great! Thanks for the advice
AI: You're most welcome. So, in what ways do your parents' expectations worry you?
H: I am scared of failing and not living up to their expectations
AI: Have you been able to explain that to them? That you're worried that you might not be able to achieve what they want you to achieve?
H: No. I have not
AI: It's certainly a difficult situation. Do you have any siblings that your parents compare you to?
H: Yes. i have a brother who is 4 years older than me
...
AI: I think it's important to remember that what you're feeling is normal and it is okay. We all want to make our parents proud. But you're only little and you can do it!
H: Thanks we can do it!
AI: I have every confidence! I hope that you have a lovely weekend!
H: I will since i have taken your advice and talk to the counselor thank you. hope you have a wonderful weekend as well

---

**Donation Solicitation.** We use the following system prompt:

---

**System Prompt - Forward Model (Donation)**

You are an expert at understanding how people think and respond when asked to donate to charities. You are able to successfully predict how real people will respond based off of only a few lines of dialogue.

---

Here is an example user prompt, using a prefix from the same dialogue shown above in the Hindsight Controller example.

---

**User Prompt - Forward Model (Donation)**

Below is a prefix of a dialogue between a charity worker ("AI:") and potential donor ("H:"). The charity worker tries to convince the potential donor to make a donation to Save the Children, the charity that the worker represents. Complete the dialogue so that the generated responses by the charity worker and potential donor are representative of what each person had said in the prefix in linguistic style and semantics. The potential donor should make a clear decision where they either donate some amount between 0 and 2 dollars or decides to not donate anything. The decision that the potential donor makes should be consistent with their behavior throughout the dialogue.

AI: Good Morning!  I am asking for a donation to Save the Children.  Have you heard of it?
H: Good morning!.. Have a nice day... Ya I heard something about it
AI: It is an International Charity to help children's lives become more stable through education, food, and good health.
H: Oh.. That's very interesting.... What is the name of Charity??
AI: Save the Children. I think that we need to help these children. It would be in the best interest of all of us to have healthy well educated children, especially in developing countries where children suffer the most. Don't you agree?
H: Ya... I agree with you... We need to take care of those children.
AI: Many children in these countries live in a war zone. Places like Syria and parts of Nigeria are so dangerous that children do not have the chance of a happy, healthy life. If you could donate a small amount from your task payment, it would be big help to these unfortunate little people.
H: Ya... I feel very bad for those children's. If those small amount will help them we should do them
AI: Your support is greatly appreciated, and no amount is too small to make a significant impact. Many donors start with just $1, which can provide a meal for a child in need. Knowing you've made such a tangible difference can be truly rewarding. How does starting with a $1 donation sound to you today? Thank you for considering it.

---

The output for this example is as below:

---

**Output - Forward Model (Donation)**

H: It's sound good... I will do that. After finishing the task I will do the donation
AI: Fantastic! Your donation will be securely deducted from your task payment. You'll be able to review the exact amount that will be deducted before submitting your responses. Thank you so much for your support!
H: Welcome... Nice to talking with you
AI: It was very nice chatting with you. Thank you for your attention and the attention you are willing to give those children.
H: Ya I will do my best for those children. Thank you... Have a nice day
AI: You are welcome! Thank you! Have a wonderful day! Bye!

---

## B.3 REWARD MODEL

Here we show the prompts we used to ask GPT-3.5 (OpenAI, 2022) to label each dialogue with the desired reward. We utilize a few-shot prompting style (Kwon et al., 2023) to ensure that reward labels are consistent with those in the original dataset.

**Counseling.** Reward in the counseling task is measured by the difference between the initial and the final intensity of negative emotion, measured on a Likert scale from 1-5. Since all of our examples are generated from prefixes of the original dialogues, we use the initial emotional intensity given in the original dataset and only label the final emotional intensity.

We use the below system prompt:

---

**System Prompt - Reward Model (Counseling)**
You are an expert at analyzing conversations between a supporter and help-seeker, where the supporter provides emotional support to the help-seeker.

---

The user prompt is shown here with a sample dialogue. The dialogues used as few-shot examples were selected at random from the original dataset.

---

**User Prompt - Reward Model (Counseling)**
Below are 10 completed dialogues between a supporter ("AI:") and a help-seeker ("H:"), who is requesting emotional support for a challenge that they are facing. Before and after each dialogue, the help-seeker rates how strong their negative emotion is on a Likert scale of 1-5 (5 being the most negative), so a lower rating for their final emotional intensity means that the supporter did a good job of addressing their problem.

<Dialogue 1>
Initial Emotional Intensity: 4
Final Emotional Intensity: 2

<Dialogue 2>
Initial Emotional Intensity: 4
Final Emotional Intensity: 1

...

<Dialogue 10>
Initial Emotional Intensity: 5
Final Emotional Intensity: 3

Lastly, here is a dialogue where the help-seeker has given their initial emotional intensity. Based on how effective the dialogue is, rate their final emotional intensity as a number between 1 to 5.

AI: Hello there, how are we doing today?
H: Hi ineed help, parents are putting to much pressure on me
AI: Im sorry to hear that your parents are putting to much pressure on you, do you want to go into further detail?
H: Yes. For some reason my parents put so much pressure on me to do well in school its killing me.
...
AI: I think it's important to remember that what you're feeling is normal and it is okay. We all want to make our parents proud. But you're only little and you can do it!
H: Thanks we can do it!
AI: I have every confidence! I hope that you have a lovely weekend!
H: I will since i have taken your advice and talk to the counselor thank you. hope you have a wonderful weekend as well
Initial Emotional Intensity: 4

What is the final emotional intensity? Give a number between 1 to 5 in the form of a line "Final Emotional Intensity: <number>". Do not provide any additional details.

**Output - Reward Model (Counseling)**
Final Emotional Intensity: 2

---

**Donation Solicitation.** The reward label for this task is based on the final donation amount. However, not all generated dialogues could be accurately labeled, either because the potential donor never specifies a donation amount, or simply because the conversation is unfinished. Thus we employed a two-step process to label the rewards: (1) check that the conversation is finished and a donation decision has been made, and only if both are true then (2) identifying the numerical donation amount.

We use the below system prompt for both calls to the model:

---

**System Prompt - Reward Model (Donation)**
You are an expert accountant who is looking through conversations for donation record keeping.

---

This is the first user prompt, in which we identify if the potential donor has decided to donate or not, with a sample dialogue. Dialogues that are deemed unfinished do not progress to the second stage and are discarded.

---

**User Prompt 1 - Reward Model (Donation)**
Below are 6 completed dialogues between a charity worker ("AI:") and a potential donor ("H:") with a label indicating if the dialogue is unfinished. In the dialogue, the charity worker tries to convince the potential donor to make a donation to Save the Children, the charity that the worker represents. The donor is usually donating a portion of the task payment of 2.0, but may donate more. In the dialogue, the potential donor should commit to donating some amount, or at least choose to not donate anything. If not, then the dialogue is unfinished.

At the end is an unlabelled dialogue also between a charity worker and potential donor. From the dialogue, identify if the dialogue is unfinished.

<Dialogue 1>
Unfinished: Yes

<Dialogue 2>
Unfinished: Yes

...

<Dialogue 5>
Unfinished: No

<Dialogue 6>
Unfinished: No

AI: Good Morning! I am asking for a donation to Save the Children. Have you heard of it?
H: Good morning!.. Have a nice day... Ya I heard something about it
AI: It is an International Charity to help children's lives become more stable through education, food, and good health.
H: Oh.. That's very interesting.... What is the name of Charity??
...
AI: Fantastic! Your donation will be securely deducted from your task payment. You'll be able to review the exact amount that will be deducted before submitting your responses. Thank you so much for your support!
H: Welcome... Nice to talking with you
AI: It was very nice chatting with you. Thank you for your attention and the attention you are willing to give those children.
H: Ya I will do my best for those children. Thank you... Have a nice day
AI: You are welcome! Thank you! Have a wonderful day! Bye!

In the dialogue, the potential donor should commit to donating some amount, or at least choose to not donate anything. If not, then the dialogue is unfinished. Is the dialogue unfinished? Answer yes/no.

**Output 1 - Reward Model (Donation)**
No

---

This is the second user prompt to label the numerical donation amount, with the same sample dialogue as above. There is a larger proportion of few-shot examples with a reward value of 0 because these dialogues were chosen such that the average reward reflects that of the original dataset.

---

**User Prompt 2 - Reward Model (Donation)**
Below are 5 completed dialogues between a charity worker ("AI:") and a potential donor ("H:") with a labelled final donation amount. In each dialogue charity worker tries to convince the potential donor to make a donation to Save the Children, the charity that the worker represents. The donor is usually donating a portion of the task payment of 2.0, but may donate more.

At the end is an unlabelled dialogue also between a charity worker and potential donor. From the dialogue, identify how much the potential donor ("H:") decides to donate to the charity (0.0 is allowed) in the form of a line "Final Donation Amount: <number>".

<Dialogue 1>
Final Donation Amount: 0.0

<Dialogue 2>
Final Donation Amount: 0.0

<Dialogue 3>
Final Donation Amount: 0.0

<Dialogue 4>
Final Donation Amount: 1.0

<Dialogue 5>
Final Donation Amount: 2.0

AI: Good Morning! I am asking for a donation to Save the Children. Have you heard of it?
H: Good morning!.. Have a nice day... Ya I heard something about it
AI: It is an International Charity to help children's lives become more stable through education, food, and good health.
H: Oh.. That's very interesting.... What is the name of Charity??
...
H: Ya I will do my best for those children. Thank you... Have a nice day
AI: You are welcome! Thank you! Have a wonderful day! Bye!

**Output 2 - Reward Model (Donation)**
Final Donation Amount: 1.0

---

## B.4   POLICY OPTIMIZATION

We use the hyperparameters reported in Table 3. All algorithms were trained on a single TPUv3 on Google Cloud until convergence. SFT took around 12 hours whereas ILQL took around 2 days until completion.

| Hyperparameter | Setting |
|---|---|
| ILQL $\tau$ | 0.8 |
| ILQL $\alpha$ | 0.0 |
| Discount factor | 0.99 |
| Batch size | 128 |
| Target network update $\alpha$ | 0.005 |
| Number of updates per iteration | 60 |
| Number of iterations | 100 |
| Optimizer | AdamW |
| Learning rate | 1e-4 |

Table 3: Hyperparameters used during training.

## B.5  USER STUDY DETAILS

In this section, we provide additional discussion on the user study used in the evaluation results in Section 5.

**Subject Allocation.**    We recruited 15 participants for our study, 10 male and 5 female with an average age of 26. 11 participants were university students, and the remaining were working in the tech industry. 9 participants have English as their native language, but all participants demonstrate fluency in English. Finally, all participants were instructed to behave as themselves and not adopt any alternative personas, and are aware that their responses were being recorded.

**Evaluation Protocol.**    During evaluation, each user is presented with a web interface where they are allowed to interact with each agent in a chat window. Each agent is anonymized, and in a random order. The user is allowed 3 minutes to familiarize themselves with the interface, and is instructed to response realistically to each agent. The user interacts with each agent in order, for a total of 3 conversations. Each conversation ends automatically after 10 turns of dialogue, but the user is allowed to end each dialogue prematurely. At the end, the users were given a Google Forms survey where they rate the performance of each agent based on the metrics listed, and on a Likert scale. We average the results across all users in Table 1 in the main paper.

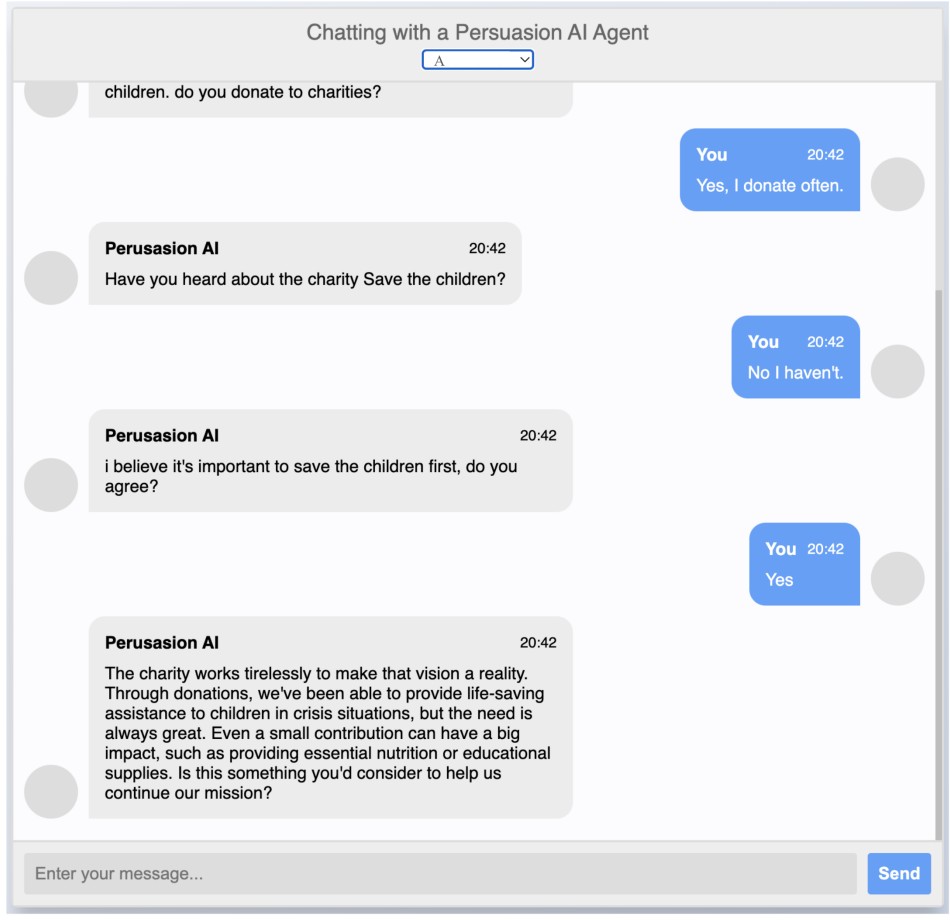

Figure 6: Chat interface used during our user study. Each agent is anonymized and in a random order that is different per user.

