# OpenReview forum: "Interactive Dialogue Agents via Reinforcement Learning with Hindsight Regenerations"
_ICLR.cc/2025/Conference — Submitted to ICLR 2025_

### Official Review · Reviewer_J8iH · 2024-11-04

**Soundness:** 2
**Presentation:** 4
**Contribution:** 2
**Rating:** 5
**Confidence:** 4

**Summary:**

The authors introduce a method to augment existing datasets with hindsight regenerations. This method uses three components (i.e., hindsight controller, forward model and reward model) and four steps (Section 4 and Figure 2 provide a good overview of the method), in short the hindsight controller proposes alternative actions (using the full dialogue), a forward model simulates completed dialogues, the reward model evaluates dialogues and the generate dialogues (plus the original once) are used as trajectories in an offline reinforcement learning to learn the final policy.

The authors compare the trained model with advanced CoT baselines, and three method and data ablations. The human evaluation results  shows improvements in two domains such Mental health support and Soliciting charitable donations

**Strengths:**

Originality
- the proposed hindsight methods is original (to the best of my knowledge) and provide an effective way to train RL based dialogue systems.

Clarity
- the overall paper is clear and easy to follow. I would have preferred more detail on the RL methodology, especially in section 4.3 ( Policy Optimization), where the authors provided a compacted explanation for a well known concept in RL but less well-known when applied to LLMs.

**Weaknesses:**

Significance & Quality
- The human evaluation lacks of rigor and details. The paper does not provide enough details on how these 15 users has been instructed nor from what demographic comes from (e.g., english proficiency, etc). This is important to evaluate the significance of the results.
- The results provide in Table 1 shows high variance, and no t-test or annotator inter-agreement is provided. Also the fact that a GPT-3 based prompting (ProCoT) has lower fluency is strange, usually LLM gets high human evaluation number.  Nevertheless this might be an artifact of a small annotator pool.
- Although not super accurate, the authors did not provide any automatic evaluation, this would have provided, even if minimal, a hook to compare different models.

**Questions:**

- How are the annotators instructed and trained for the task?
- What is the statistical significance of the results in Table 1?

---

> ### Author Response · Authors · 2024-11-22
> **Author Response to Reviewer J8iH**
>
> Thank you for your review. You raised several important concerns that we aim to address.
>
> **User study details**
>
> This is an excellent point. We have updated the Appendix of our paper with details on the demographics of the user study, as well as specifics on what they were instructed to do and what their interface looked like (see Appendix B.5).
>
> On a high-level, we recruited $15$ participants for our study, $10$ male and $5$ female with an average age of $26$. $11$ participants were university students, and the remaining were working in the tech industry. $9$ participants have English as their native language, but all participants demonstrate fluency in English. Finally, all participants were instructed to behave as themselves and not adopt any alternative personas, and are aware that their responses were being recorded.
>
>
> **Automatic evaluation and noisy results**
>
> We agree that most of our metrics come from a user study, which can be noisy and potentially lack objectivity. However, to address this, we do additionally measure reward obtained from simulating a much larger number of users (as seen in Reward (sim) row in Table 1). We believe such a metric does not suffer from lack of objectivity or noisiness that our user study metrics may have. As shown in Table 1, such a metric is derived from automatically simulating $400$ users and shows a more statistically convincing improvement in performance.  If you have any other suggestions on what can be measured in our evaluations, we would be happy to do so!

---

> ### Author Response · Authors · 2024-12-02
> **Let us know if you have any further questions!**
>
> As the discussion period is about to end soon, please let us know if you have any further questions or concerns, and we would be happy to address them!

---

### Official Review · Reviewer_YxT3 · 2024-11-04

**Soundness:** 2
**Presentation:** 2
**Contribution:** 2
**Rating:** 3
**Confidence:** 4

**Summary:**

The paper explores training dialogue agents that use reinforcement learning (RL) to improve their interactive capabilities, especially in complex tasks like mental health support and charitable persuasion. Unlike standard models focused on one-off responses, the authors propose a method called hindsight regeneration. This approach involves augmenting existing dialogue datasets by having the model review conversations in hindsight, identifying optimal actions post-interaction, and suggesting better responses to refine the dataset. The method employs offline RL, avoiding costly real-time exploration, and allows agents to generate more effective strategies over multiple dialogue turns.

**Strengths:**

By refining conversations after they occur, the model can learn from suboptimal dialogues and incorporate more effective conversational strategies. This approach allows for the development of more adaptive and goal-directed dialogue agents capable of managing complex, multi-turn interactions.

**Weaknesses:**

The main contribution of this paper seems to be only a dataset construction method, without any innovations in model or methodology. Since this is not a dataset track paper, the authors appear to be primarily reporting experimental results rather than providing a substantial contribution. If the authors can convince me otherwise, I would be willing to adjust my score.

**Questions:**

1. Although the author believes that BLEU and ROUGE metrics may not fully capture the model’s performance in dialogue, including these metrics could strengthen the experimental results.

2. It seems the author did not specify the datasets used for SFT and RL training.

3. The author introduces a human evaluation conducted by a group of 15 users. Could the author provide details on these users’ backgrounds, educational levels, and native languages? Additionally, was any consistency check performed on the scores given by these 15 individuals?

4. The author’s experiments use only LLaMA-7B as the base model. To demonstrate the generalizability of the approach, has the author considered adding other base models? For reference, here are a few that could be selectively added: LLaMA2-7B, LLaMA2-13B, LLaMA3-8B, Vicuna, WizardLLM, ChatGLM, MiniCPM, Qwen.

5. In Section 5, the author states, “Note that we train on a much smaller model than used in the prompting baselines, yet as we will show later, we still are able to outperform such more sophisticated LLMs.” Given that the model size for ChatGPT-3.5 has never been disclosed and the model itself has gone through several iterations, this statement may risk overclaiming.

---

> ### Author Response · Authors · 2024-11-22
> **Author Response to Reviewer YxT3**
>
> Thank you for your review. We will address your concerns with our work below. Please let us know if you have any additional questions or concerns after reading our rebuttal!
>
> **Objective metrics**
>
> We agree that most of our metrics come from a user study, which can be noisy and potentially lack objectivity. We do not view BLEU or ROUGE as suitable metrics because in our domains, it is apparent that the data in demonstrations exhibits very suboptimal behavior. Thus, BLEU or ROUGE, which measure how well our approach mimics the demonstrations, would not be an appropriate evaluation metric. However, we do additionally measure reward obtained from simulating a much larger number of users (as seen in Reward (sim) row in Table 1). We believe such a metric does not suffer from lack of objectivity or noisiness that our user study metrics may have. If you have any other suggestions on what can be measured in our evaluations, we would be happy to do so!
>
> **Specify datasets for SFT and RL**
>
> Apologies if we made this unclear in our work. All the non-prompting baselines use the same dataset as our approach, and are meant to act as ablations to our method. We have made this clearer in our updated paper.
>
> **User study details**
>
> This is an excellent point. We have updated the Appendix of our paper with details on the demographics of the user study, as well as specifics on what they were instructed to do and what their interface looked like (see Appendix B.5).
>
> On a high-level, we recruited $15$ participants for our study, $10$ male and $5$ female with an average age of $26$. $11$ participants were university students, and the remaining were working in the tech industry. $9$ participants have English as their native language, but all participants demonstrate fluency in English. Finally, all participants were instructed to behave as themselves and not adopt any alternative personas, and are aware that their responses were being recorded.
>
> **Adding more base models**
>
> This is another great point. We chose LLama-7B because it was the largest LLM where we could train a policy using a reasonable amount of compute. Our goal was to consider the peak potential of each evaluated method, given our limited compute resources. We believe that our approach would only work better and with less data if we were to train on a larger, more sophisticated base LLM.

---

> ### Author Response · Authors · 2024-12-02
> **Let us know if you have any further questions!**
>
> As the discussion period is about to end soon, please let us know if you have any further questions or concerns, and we would be happy to address them!

---

### Official Review · Reviewer_X57i · 2024-11-04

**Soundness:** 2
**Presentation:** 2
**Contribution:** 2
**Rating:** 5
**Confidence:** 4

**Summary:**

This paper introduces a method for enhancing dialogue systems using pre-trained language models (LLMs) through hindsight reinforcement learning (RL).
It utilizes LLMs to simulate human responses and generate new dialogue samples, which are then used to improve or supplement the original dataset, addressing the problem of lacking effective strategies in dialogue systems. The authors conducted experiments on two challenging tasks and demonstrated the effectiveness of this method. Additionally, they showed how to adjust dialogue strategies based on user feedback to achieve more natural and effective conversational interactions.

**Strengths:**

1. The article proposes a data augmentation method to improve the performance of dialogue agents.

2. It significantly outperforms existing fine-tuning methods in terms of efficiency, naturalness, and usefulness.

**Weaknesses:**

1. The article's innovation is limited.

2. The introduction of the article's methodology is not sufficiently clear.

3. Table 1 is too long.

4. The experimental analysis is insufficient.

5. There is a lack of objective evaluation metrics.

**Questions:**

NA.

---

> ### Author Response · Authors · 2024-11-22
> **Author Response to Reviewer X57i**
>
> Thank you for your review. We would like to address the weaknesses you identified with our paper.
>
> **Innovation is limited**
>
> We agree that our method is relatively straightforward, and draws inspiration from many model-based RL methods in existing RL literature such as Dyna and MBPO. However, as far as we are aware, we are the first to propose such an algorithm using LLMs rather than separately learning an environment model, and achieve strong results in multiple dialogue domains. Hence, we do not believe the simplicity of the approach should be seen as a downside.
>
> **Experimental analysis is insufficient**
>
> We would appreciate it if you would be willing to expand on what is lacking in our evaluations. We believe we consider all the relevant state-of-the-art baselines and ablations of our method, and in two challenging domains. You raise a good point about objective metrics, as most of our results come from a user study. However, we additionally measure reward obtained from simulating a much larger number of users (as seen in Reward (sim) row in Table 1). We believe such a metric does not suffer from lack of objectivity or noisiness that our user study metrics may have.

---

> ### Author Response · Authors · 2024-12-02
> **Let us know if you have any further questions!**
>
> As the discussion period is about to end soon, please let us know if you have any further questions or concerns, and we would be happy to address them!

---

### Official Review · Reviewer_eQXb · 2024-11-05

**Soundness:** 3
**Presentation:** 3
**Contribution:** 3
**Rating:** 6
**Confidence:** 3

**Summary:**

The paper looks at training an LLM to play the role of a conversational agent, particularly in situations which require some manipulation for lack of a better term of the user towards a goal, for example making a donation to charity.

The paper's main idea is that rather than looking at conventional RL based exploration guided only by a numeric reward function, to look at generated dialogs in full and to with "hindsight" identify turns where the dialog may be judged as having been suboptimal and steered the user away rather than towards the desired goal. Such data can then be regenerated from that sub-optimal turn forwards. This is done and that data is then used in an offline RL algorithm to update the LLM playing the role of the system.

The paper compares to some prior works and reports results on 2 datasets.

**Strengths:**

The observation that feedback is more precise at the level of dialog regenerations having seen a full "rollout" is valid, given reward based feedback alone is a more difficult search space to optimise.

The paper is clear in its motivations, compares against other published works and reports reasonable results based on an evaluation with different users.

**Weaknesses:**

One minor suggestion: it's claimed that the results in table 1 are statistically significant (line 453) however there's no detail given for how this was determined. This should be included.

**Questions:**

* Would doing RLHF/DPO on the bad versus better turn be a valid comparison to include here? By "bad" I mean the turn first identified as being sub-optimal in the hindsight stage, and by "good" I mean it's regenerated version. Unclear how similar this is to the offline ILQL algorithm which I admit to not looking up now.

* what happens if you do multiple rounds of hindsight re-generation?

---

> ### Author Response · Authors · 2024-11-22
> **Author Response to Reviewer eQXb**
>
> Thank you for your review. You raised several questions that we aim to address below.
>
> **RLHF/DPO baseline**
>
> You raise a good point that RLHF/DPO can be used on the original vs regenerated utterances to potentially improve the policy. We see one major downside with such an approach. Namely, rather than training on entire dialogues, we would be training on pairs of utterances (one per dialogue), which contains much less information. This may not be an issue if the base model is already sophisticated enough, but since state-of-the-art LLMs do not expose APIs for such training, we did not  find this to be the case. We made some preliminary experiments using DPO on the LLama-7B model, which is the same model used to train our methods, and found it performed significantly worse in simulation, achieving $0.36 \pm 0.17$ reward (this can be directly compared with the Reward (sim) row in Table 1). However, we believe this is still an interesting baseline and will add it to the main paper.
>
> **Multiple rounds of hindsight iteration**
>
> This is also an interesting point, which could lead to different behaviors. The reason why we choose not to do so is that we believe biases can arise in the forward generation step where a new dialogue is simulated. In particular, instruction fine-tuned LLMs tend to generate responses with a positive or agreeable bias, which would be exacerbated by multiple rounds of iteration.

---

> ### Author Response · Authors · 2024-12-02
> **Let us know if you have any further questions!**
>
> As the discussion period is about to end soon, please let us know if you have any further questions or concerns, and we would be happy to address them!

---

### Meta-Review · Area_Chair_aj3N · 2024-12-19

**Metareview:**

The main contribution of the paper is in realizing that LLMs can self diagnose their mistakes posthoc once they see how a human responds to them. This enables them to rewrite trajectories and then train on them in a manner reminiscent of such hindsight methods in deep RL in other domains. That said, most of the reviewers do not find the experimental methodology convincing to prove that this method provides significant improvements over alternatives. I do not believe this paper is ready for publication in its current form.

**Additional Comments On Reviewer Discussion:**

I am discounting the review of X57i as it is very minimal and the critiques unsubstantiated. The reviews by the remaining reviewers are sound though seemingly not fully addressed during my read of the discussion. None of the reviewers followed up after the authors responded.

---

### Decision · Program_Chairs · 2025-01-22

Reject